# Haemoparasitic Infections in Cattle from a *Trypanosoma brucei* Rhodesiense Sleeping Sickness Endemic District of Eastern Uganda

**DOI:** 10.3390/tropicalmed5010024

**Published:** 2020-02-07

**Authors:** Enock Matovu, Claire Mack Mugasa, Peter Waiswa, Annah Kitibwa, Alex Boobo, Joseph Mathu Ndung’u

**Affiliations:** 1College of Veterinary Medicine, Animal Resources and Biosecurity, Makerere University Kampala, P.O. Box 7062 Kampala, Uganda; matovue@covab.mak.ac.ug (E.M.); pwaiswa@covab.mak.ac.ug (P.W.); kitty.ann34@yahoo.com (A.K.); alexboobo@yahoo.com (A.B.); 2Foundation for Innovative New Diagnostics, Campus Biotech, Chemin des Mines 9, CH 1202 Geneva, Switzerland; joseph.ndungu@finddx.org

**Keywords:** Haemoparasites, human African trypanosomiasis, elimination, animal reservoirs

## Abstract

We carried out a baseline survey of cattle in Kaberamaido district, in the context of controlling the domestic animal reservoir of *Trypanosoma brucei rhodesiense* human African trypanosomiasis (rHAT) towards elimination. Cattle blood was subjected to capillary tube centrifugation followed by measurement of the packed cell volume (PCV) and examination of the buffy coat area for motile trypanosomes. Trypanosomes were detected in 561 (21.4%) out of 2621 cattle screened by microscopy. These 561 in addition to 724 apparently trypanosome negative samples with low PCVs (≤25%) were transported to the laboratory and tested by PCR targeting the trypanosomal Internal Transcribed Spacer (ITS-1) as well as suspect Tick-Borne Diseases (TBDs) including Anaplasmamosis, Babesiosis, and Theileriosis. PCR for *Anaplasma* sp yielded the highest number of positive animals (45.2%), followed by *Trypanosoma* sp (44%), *Theileria* sp (42.4%) and *Babesia* (26.3%); multiple infections were a common occurrence. Interestingly, 373 (29%) of these cattle with low PCVs were negative by PCR, pointing to other possible causes of aneamia, such as helminthiasis. Among the trypanosome infections classified as *T. brucei* by ITS-PCR, 5.5% were positive by SRA PCR, and were, therefore, confirmed as *T. b. rhodesiense.* Efforts against HAT should therefore consider packages that address a range of conditions. This may enhance acceptability and participation of livestock keepers in programs to eliminate this important but neglected tropical disease. In addition, we demonstrated that cattle remain an eminent reservoir for *T. b. rhodesiense* in eastern Uganda, which must be addressed to sustain HAT elimination.

## 1. Introduction

African trypanosomes transmitted by tsetse flies (*Glossina* sp.) cause the zoonotic human African trypanosomiasis (HAT; also known as sleeping sickness) as well as animal African trypanosomiasis (AAT; nagana). AAT is a major hindrance to livestock productivity in tsetse infested areas of sub-Saharan Africa. This disease was reported to affect various animal productivity parameters, including growth, mortality, calving rate, draft power, meat, and milk production by up to 20% in susceptible animals [1]. The economic losses attributable to AAT were estimated at US$4.5bn per annum [2]. On the other hand, the human disease (HAT) was for many years among the leading causes of death in rural areas. HAT is caused by two subspecies of *T. brucei* that are able to resist the naturally occurring trypanolytic factor (APOL I) and establish infections in humans. The chronic form of HAT associated with *T. b. gambiense* (gHAT) occurs in central and western Africa (including parts of northwestern Uganda), while the acute *T. b. rhodesiense* (rHAT) is found in eastern and southern Africa, in a belt presently stretching from eastern Uganda through Tanzania to Malawi and Zambia. In the past 5 decades, the number of HAT cases ranged between 50,000 and 70,000, dropped to below 10,000 in 2009, and continued to drop to 6743 cases by 2011 [3]. This reduction in HAT incidence was as a result of campaigns spearheaded by the World Health Organization (WHO) working together with non-governmental organizations [4] as well as National control programs. Consequently, in 2012, the WHO included HAT on the list of diseases set for elimination, first as a public health problem by 2020 followed by complete interruption of transmission by 2030 [5].

The role of animal reservoirs in rHAT transmission was recognised by pioneer researchers [6,7] and was the basis for game destruction as a method of sleeping sickness control during colonial times. In Uganda, research carried out during the 1980s and 1990s singled out cattle, pigs and dogs as the domestic animal reservoirs of rHAT [8,9,10,11]. According to Simarro et al. (2010) [12], eastern Uganda contributed over 50% of *T. b. rhodesiense* reported cases in Africa between 2000 and 2009; many of these were of livestock reservoir origin. Indeed, the latest outbreak that spilled over to Teso and Lango regions was attributed to cattle movement from the southerly endemic areas [13,14]. In line with the above facts, Uganda embraces a control strategy that involves surveillance and treatment of all detected HAT cases, vector control to supress tsetse populations, thereby limiting transmission, as well as control of the animal reservoir by chemotherapy. However, full implementation of this strategy is hampered by limited resource availability such that some aspects cannot be consistently executed.

In this baseline survey to support elimination of HAT by targeting the animal reservoir, we aimed to identify the major haemoparasites particularly associated with the typically low packed cell volume (PCVs) of less or equal to 25% of cattle blood observed both in the presence and absence of the motile trypanosomes in buffy coats following capillary tube centrifugation.

## 2. Materials and Methods

### 2.1. Study Area and Study Population

The livestock survey was carried out in Kaberamaido district (approximate latitudes 1.5500 to 2.3834 and longitudes 30.0167 to 34.3000), in Eastern Uganda (Figure 1). A total of 15 parishes (Ochuloi, Opilitok, Kaberikole, Omoru, Amukurat, Anyara, Kalaki, Ariamo, Abalang, Palatau, Achan-pii, Kamuk, Omarai, Aperkina and Abalkweru) in five sub-counties (Otuboi, Kalaki, Alwa, Kaberamaido and Kobulubulu) were included in the survey. The main occupation in the entire study area is subsistence agriculture. Cattle are the major livestock and communal grazing is usually practiced.

### 2.2. Study Design and Field Surveys

This was a cross-sectional study carried out in the above-mentioned parishes. Cattle were screened by a mobile team at a designated site per parish, selected with assistance from the District Veterinary Officer (DVO) and local leaders. Cattle blood was drawn by venipuncture into EDTA-coated vacutainer tubes and subjected to Haematocrit Centrifugation Technique (HCT) [15]. Packed cell volume (PCV) readings were taken for all the samples using a manual micro-haematocrit reader (Thomas Scientific, Swedesboro, NJ, USA) [16]. This was followed by examination of the buffy coat area of the centrifuged capillary tubes under the microscope (Leica Microsystems, Wetzlar, Germany) at 100× magnification for the presence of motile trypanosomes. Aliquots of blood from cattle with low PCVs, regardless of whether they had detectable trypanosomes or not, were transported in liquid nitrogen to the laboratory for further analysis. In addition, representative thin smears from apparently trypanosome-negative samples were made and fixed with methanol, then stored in slide boxes and transported to the laboratory in slide boxes for staining and examination for possible presence of other haemoparasites.

The animals were treated with either diminazene aceturate (3.5 mg/kg body weight) or isometamidium chloride (1 mg/kg body weight) as per design of the mother project for which this was the baseline study. In addition, deltamethrin pour-on as a tsetse control tool was applied superficially on all the cattle to control tsetse flies.

## 3. Laboratory Procedures

### 3.1. Staining of Blood Smears

The thin blood smears were stained with acridine orange following protocols developed by the Foundation for Innovative New diagnostics (FIND) [17] and examined for the presence of tick-borne pathogens under a fluorescence microscope (Carl Zeiss Microscopy, Jena, Germany) at 400× magnification.

### 3.2. Extraction of Genomic DNA

Genomic DNA was extracted from 100 μL of whole blood samples using a commercial Quick gDNA mini prep kit (Zymo Research, Irvine, CA, USA) following the manufacturer’s instructions. It was eluted in 50 μL PCR water and stored frozen at −20 °C until use in the PCR reactions.

### 3.3. Identification Trypanosome Species by PCR

All PCRs in this study were done using the My Taq Mix^®^ (Bioline, London, UK) (https://cn.bioline.com/mytaq), while primers were ordered from Microsynth company (The Swiss DNA company, Bern, Switzerland).

The infecting trypanosome species were identified by PCR using primers targeting the Internal Transcribed Spacer-1 (ITS-1) region of the rDNA as described by Njiru et al. [18]. PCR reactions were performed in a volume of 25 µL, containing 1x My Taq Mix polymerase enzyme, the primer pair ITS1-CF (5’ CCG GAA GTT CAC CGA TAT TG 3’) and ITS1-BR (5’ TTG CTG CGT TCT TCA ACG AA 3’) each at 0.5 µM. Amplification was performed under the following conditions; 94 °C for 5 min (initial denaturation) followed by 35 cycles of 94 °C, 1 min (denaturation), 60 °C, 1 min (annealing), 72 °C, 1 min (extension) and a final extension of 72 °C for 5 min. Three microlitres of genomic DNA was added to each PCR reaction as template. A positive control (*Trypanosoma brucei brucei* GVR-35 strain) and a negative control (double distilled water) were included alongside the test samples. Ten microlitres of each amplicon was subjected to electrophoresis on a 2% agarose gel containing ethidium bromide (0.5 µg/mL). The amplified products were visualized using an ultra violet transilluminator (Waghtech international) and the band sizes estimated by comparison with a standard DNA marker (www.Finnzymes.com).

The DNA samples that were negative with the single step ITS-PCR described above were thereafter subjected to nested ITS-PCR as described by Cox et al. [19] to rule out negativity due to limiting quantities of trypanosomal DNA in the samples. The primary PCR reaction mixture was 25 µL total volume, with the primer pair ITS-1 (5’ GAT TAC GTC CCT GCC ATT TG 3’) and ITS-2 (5’ TTG TTC GCT ATC GGT CTT CC 3’) each at 0.5 µM. Five microlitres of genomic DNA was added to each PCR reaction as template and amplification cycle included 98 °C for 1 min (initial denaturation) followed by 25 cycles of 98 °C, 5 s (denaturation), 64 °C, 30 s (annealing), 72 °C, 30 s (extension) and a final extension of 72 °C for 10 min. This was followed by the second PCR reaction where the primer pair ITS3 (5’ GGA AGC AAA AGT CGT AAC AAG G 3’) and ITS4 (5’ TGT TTT CTT TTC CTC CGC TG 3’) each at 0.5 µM concentration and 5 µL of primary PCR product as the template were used. The amplification was performed under similar conditions, with controls included as above. Gel electrophoresis was done in 1% agarose alongside a 1 kb standard DNA size marker (Bioline, London, UK).

### 3.4. Identification of Trypanosoma brucei Bub-Species by PCR

The DNA samples that were positive for *Trypanosoma brucei* species by single step or nested ITS-PCR were also subjected to a nested PCR using sub-species-specific primers that target the Serum Resistance Associated (SRA) gene [20] that is specific to *T. b. rhodesiense* (*T.b.r*). In the first run, amplification of three microlitres of template DNA was performed in a 25 µL reaction volume with the primer pair, SRA outer-s 5’ CCT GAT AAA ACA AGT ATC GGC AGC AA 3’ and SRA outer-as 5’ CGG TGA CCA ATT CAT CTG CTG CTG TT 3’ each at 0.5 µM concentration. The thermocycling conditions were as follows; 98 °C for 1 min (initial denaturation) followed by 25 cycles of 98 °C, 5 s (denaturation), 64 °C, 30 s (annealing), 72 °C, 2 min (extension) and a final extension of 72 °C for 1 min. In the second run, three microlitres of product from first run was amplified under similar conditions as in the first run but using primer pair, SRA inner-s 5’ ATA GTG ACA TGC GTA CTC AAC GC 3’ and SRA inner-as 5’ AAT GTG TTC GAG TAC TTC GGT CAC GCT 3’ also at 0.5 µM. A negative control, double distilled water (with no template DNA added) and a positive control, *T.b.r* 729 strain (Molecular biology laboratory, MUK-COVAB) were included in the PCR amplification. Electrophoresis was done in 2% agarose gels.

### 3.5. PCR Amplification for Anaplasma Species

In the first screen of stained blood smears, *Anaplama* sp., *Babesia* sp., and *Theileria* sp. were detected in some of the slides. This informed us of the choice tick-borne parasites to screen for in the entire set of low PCV samples using previously published specific PCRs.

PCR amplification for *Anaplasma* species was with specific primers targeting the 16S rRNA [21]. The thermo cycling profile was 95 °C for 5 min (initial denaturation) followed by 45 cycles of 95 °C, 30 s (denaturation), 51 °C, 30 s (annealing), 72 °C, 45 s (extension) and a final extension of 72 °C for 10 min. A positive control, (bovine field isolate confirmed with *Anaplasma* species) and a negative control (double distilled water with no template DNA added) were included in the PCR amplification; electrophoresis was in a 1.5% agarose gel alongside a 100 bp standard DNA marker (Bioline, London, UK).

### 3.6. PCR Amplification for Babesia Species

For *Babesia* species, PCR using a primer pair that target 18S rRNA gene [22] was performed in 25 µL PCR mixture containing 5 µL of DNA template, and the primer pair Bab-1s 5’ CAA GAC AAA AGT CTG CTT GAA AC 3’ and Bab-s 5’ GTT TCT GAC CCA TCA GCT TGA C 3’. Amplification was under the following conditions; 95 °C for 5 min followed by 45 cycles of 94 °C, 30 s (denaturation), 63 °C, 30 s (annealing), 72 °C, 45 s (extension) and a final extension of 72 °C for 10 min. A positive control, (bovine field isolate confirmed with *Babesia* species) and a negative control (double distilled water with no template DNA added) were included in the PCR amplification. Electrophoresis was in a 1.5% agarose gel alongside a 100 bp standard DNA marker (Bioline, London, UK).

### 3.7. PCR Amplification for Theileria Species 

Amplification was carried out by PCR targeting the small subunit (SSU) rRNA which is common to *Theileria* species [23]. All PCR reactions were performed in a volume of 25 µL with the primer pair, F (989) 5’ AGT TTC TGA CCT ATC AG 3’ and R (990) 5’ TTG CCT TAA ACT TCC TTG 3’ each at 3.2 µM. Five microlitres of genomic DNA was added to each PCR reaction as the template. The PCR conditions were 95 °C for 5 min (initial denaturation) followed by 35 cycles of 94 °C, 1 min (denaturation), 60 °C, 1 min (annealing), 72 °C, 1 min (extension) and a final extension of 72 °C for 10 min. A negative control (double distilled water with no template DNA added) and a positive control (bovine field isolate confirmed with *Theileria* species) were included in the PCR amplification. Ten microlitres of each amplicon was subjected to electrophoresis in a 1.5% agarose gel alongside a 100 bp standard DNA marker (Bioline, London, UK).

## 4. Results

In all, 2621 cattle were screened using the HCT, of which 561 were positive for trypanosomes, translating into a parasitological prevalence of 21.4%. We took representative smears from HCT negative, low PCV (≤25%) cattle for iLED microscopy to look for trypanosomes and other haemoparasites. This was in order to determine which hemoparasites to search for by PCR executed on the entire collection, in addition to *Trypanosoma* species. None of these smears had detectable trypanosomes but we identified *Babesia* sp., *Anaplasma* sp. and *Theileria* sp. DNA was therefore prepared from the 1285 low PCV samples to perform species specific PCRs for *Trypanosoma* sp., *Babesia* sp., *Anaplasma* sp. and *Theileria* sp., (Appendix A) to show to what extent each might have contributed to the low PCVs.

Of the 561 HCT positive cattle as well as 724 cattle with no detectable trypanosomes but with low PCV, trypanosomal ITS-PCR was positive in 473 and 94 samples respectively, consequently missing 15.7% of the samples in which trypanosomes had been detected by microscopy. The ITS-PCR results are summarized in Table 1. *T. brucei* was the most abundant species, present in 254 of the 567 (44.8%) positive samples, followed by *T. congolense* (38.1%), the benign trypanosome *T. theileri* (22.6%) and *T. vivax* (20.3%). In this analysis, 14 out of 254 *T. brucei* positive cattle (5.5%) were SRA positive, indicating that they were the human infective *T. b. rhodesiense*.

Considering the low PCV animals (1285, of which 561 were HCT positive and 724 negative; Table 1), and combining positive results from both HCT and ITS-PCR (composite reference; total 655 cases), 51% of these cattle had trypanosomiasis, and as such, the anaemia in 49% of them could have been associated with other causes.

Of the 567 ITS-PCR positive samples 146 (25.7%) were infected with more than one *Trypanosoma* species, the majority with two species, mainly *T. brucei* and *T. congolense* as shown in Table 2.

Considering trypanosomiasis in relation to the tick-borne haemoparasites whose PCRs were done in this analysis (Table 3), it was revealed that 492 of the 1285 low PCV cattle (38.3%) had both trypanosomes and any of the tick-borne haemoparasites. Infection with *Anaplasma* was highest (45.2%) among the cattle with low PCV, followed by *Theileria* (42.4%), and least was *Babesia* infection that accounted for 26.3%. Co-infection of trypanosomes with *Babesia* occurred in 19.5% of animals with low PCV.

Finally, from Table 3, it is noteworthy that among tested cattle with low PCV (n = 1285), no trypanosomes or any of the 3 tick-borne infections were detected in 373 cattle blood samples (29%).

## 5. Discussion

Despite its continued decline in incidence over the past decade, rHAT remains an important disease, with potential to re-emerge if relevant control measures are not sustained. rHAT is a zoonotic disease involving mainly cattle and wild animals in the transmission cycle; therefore, its control requires a multi-sectoral approach. This study aimed to identify the major haemoparasites affecting cattle in Kaberamaido district as a basis to devise appropriate strategies to accelerate and sustain elimination of rHAT. Kaberamaido is one of the districts in the cattle corridor in eastern Uganda where the latest rHAT outbreak in Uganda occurred since 2005 [24]. Since then, over 500 cases were treated at Lwala hospital alone, which serves the Kaberamaido-Lango focus. The outbreak was attributed to influx of cattle infected with *T. b. rhodesiense* from active rHAT foci in the south [13,25]. Interventions, including tsetse control and mass treatment of cattle, led to a decline in incidence recorded since the late 2000s [26].

Haemoparasitic infections have globally been documented to cause immense production losses in the livestock sector [27,28]. In Uganda, Rubaire-Akiiki et al. [29] reported the sero-prevalence of *Theileria parva* among communally grazed cattle in low lands to be as high as 70% while those of *Babesia* and *Anaplasma* were 65% and 15%, respectively. Later, in 2011, Angwech et al. [30] assessed the prevalence of tick-borne parasites in relation to different livestock production systems in Gulu district in northern Uganda and found that the prevalence of *Theileria* was highest in cattle (28.1%), that of *Anaplasma* was highest in goats (19.0%), while the prevalence of *Babesia* was highest in sheep (3.64%) under the open grazing system [30]. In yet another study conducted in central and western Uganda, the prevalence of haemoparasites was reportedly 47.4%, 6.7%, 1.9% and 14.4% for *Theileria parva*, *Babesia* spp., *Trypanasoma brucei*, *Anaplasma* spp respectively. Generally, previous studies ahave shown that livestock that are grazed openly have high prevalence rates of haemoparasites [29,30,31,32]. Thus, even in this study, which was addressing the animal reservoir of rHAT, it was important to identify other hemopasitic challenges in the district, in order to consider appropriate interventions.

In this study, a considerable proportion of trypanosome-positive samples (15.7%) were not detected by PCR. This could partly have been due to the loss of DNA quality during field collection or processing of the blood samples, a usual challenge of molecular investigations. It is also plausible that the undetected samples could have been with triple and quadruple trypanosome species infections, in which case the ITS PCR has a markedly low sensitivity, as was previously reported by Njiru et al. [33]. The scenario of double infection is common in animals, as is shown in this study as well as by Mugittu et al. [34]; however, triple and quadruple trypanosome infections, although rare, occur in animals as well as the tsetse fly vector [33,35]. Another possible explanation is that the undetected samples could have been infected with *T. vivax* strain variants that have changes in regions where the primers anneal; as was observed by Njiru et al., [18]. In that study, field samples from Kenya were analyzed using ITS-1 and gave differing sizes (250, 249, 248 base pairs) of the ITS region. Similarly, Malele et al. [36], while analyzing tsetse flies in Tanzania, reported such variation in *T. vivax* strains. Indeed, earlier in 2001, it was reported that the evolution rate of the 18S rRNA gene of *T. vivax* was significantly faster than that of other trypanosomes and specifically evolved 7 to 10 times that of non-salivarian trypanosomes [37]. Because of these changes, the primer annealing capacity may be compromised, thus the false negative results in the current study.

The current study demonstrated that only 4.5% (58) of the 1285 cattle with low PCV were infected with trypanosomes alone, 38.2% (492) had both trypanosomes and tick-borne parasites, while 29% (375 cattle) were infected with the latter in the absence of trypanosomes. Thus, based on these results, any intervention targeting trypanosomes alone would benefit less than half of the anaemic animals. This could conceal the benefits of block trypanocidal treatment campaigns from the point of view of general improvement in herd health. The implication of this is that livestock farmers might need to see a considerable improvement of herd health in order to appreciate and fully participate in control operations.

Another observation in the current study that might be of importance to policy is that 250 of the 655 composite reference positive animals (38.2%) had both trypanosomiasis and babesiosis. Thus, it might be a tough decision to make in such a scenario, whether to use isometamidum that clears the trypanosomes and offer protection for 3 months or to use diminazene aceturate that clears both parasites (trypanosomes and *Babesia*) and could lead to better (short-term) improvement in general herd health but offers no prophylaxis against trypanosomiasis [38]. The latter may call for more frequent interventions, translating into more cost and time inputs. These arguments all point to the need to fully analyse the situation and formulate relevant interventions that are likely to be readily acceptable to the animal owners, while maintaining the rational use of the trypanocides to delay the emergence of drug resistance.

The role of social science will be very crucial in this era of near-to-complete elimination or HAT, as we need innovative ways to sustain the gains accrued from the recent outbreak “fire-fighting” situations. It is clear that we need to go to the field with a more open mind and approach since there are many other challenges than trypanosomiasis alone, even though we primarily move in with rHAT control objectives. For example, the chemicals to consider for animal bait tsetse control should be those that equally affect ticks so that the pastoralists get maximum benefit from the intervention. Similarly, restricted application of insecticide to cattle [39], though indisputable with regard to effective tsetse control, should be carefully designed in order not to leave some equally important tick-borne diseases co-existing in the control areas unattended to.

Of the cattle that were positive for the trypanosomal ITS-PCR, 44.8% were infected with *T. brucei* while 5.5% of these had the human infective *T. b. rhodesiense* circulating in the animal reservoir. The significance here is that since *T. brucei* is not the most pathogenic species to cattle in absence of harsh environmental conditions such as droughts, the clinical presentation might not be so striking, to the extent that the farmers may fail to seek veterinary attention for apparently healthy looking animals. Therefore, these clinically healthy animals may continue to harbor human infective trypanosomes for long without raising suspicion. This scenario poses eminent challenges in the control of sleeping sickness in livestock farming communities, such as in this study area, and may require regular testing and treatment of the cattle reservoirs irrespective of their clinical status. To our observation, *T. brucei* tends to dominate cattle infections in active rHAT foci.

In addition to the known pathogenic trypanosomes detected in the study area, we also demonstrated the presence of the benign *T. theileri*. This implies that the nuisance biting flies are active in the area, adding to the livestock productivity constraints faced by the livestock farmers.

As outlined above, samples from cattle with low PCV were analyzed using PCR to detect DNA of trypanosomes and any of three tick-borne parasites (*Babesia*, *Theileria* and *Anaplasma*); however, 29% of these animals were negative for any of these infections. Thus, we could not attribute the low PCV values to trypanosomiasis or any of the three TBDs tested for. We suggest that other contributors to the low PCV could include helminth infections that are common in the field where no control measures are practiced, as is the case among many subsistence livestock farmers. It is thus equally important to control helminths for maximum livestock production. In other words, there might be need for a complete package to deliver to the communities in order to sustainably control rHAT; again, the important role of social scientists or social economists cannot be ignored.

## 6. Conclusions

This study revealed various haemoparasites infecting cattle in Kaberamaido, including *Theileria*, *Babesia*, *Anaplasma* and *Trypanosoma*, and suggests that trypanosomiasis (though high on the list) might not necessarily be the number one problem faced by the livestock farmers. Noteworthily, without a vigorous community engagement and education campaign, the farmers might fail to fully appreciate the contribution of domestic animals to rHAT transmission. Any rHAT elimination effort should therefore come as a package that not only secures human health but leaves behind a population with better livelihoods and economic empowerment arising from improved animal productivity. Multi-sectoral, multi- and trans-disciplinary teams shall definitely be required to address and sustain rHAT elimination. We perhaps presently need social scientists more than ever before, in the face of diminishing rHAT incidence, bearing in mind that a resurgence can happen if no properly thought out measures against this zoonosis are implemented. The notable presence of *T. b. rhodesiense* in cattle in this area reminds us that the domestic animal reservoir is still around and should be sustainably addressed.

## Figures and Tables

**Figure 1 tropicalmed-05-00024-f001:**
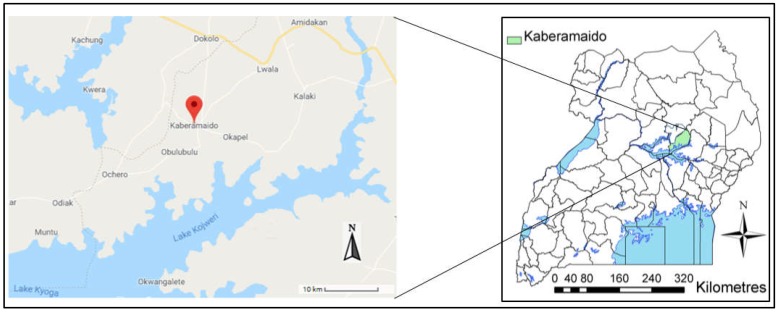
Regional map of Uganda showing the location of the study area. (https://www.google.com/maps/place/Kaberamaido).

**Table 1 tropicalmed-05-00024-t001:** ITS1-PCR results and the identified trypanosome species. Samples classified as trypanosmoe positive or negative by the HCT were tested by PCR to detect and identify the respective trypanosome species.

Infection Status	Total	ITS +ve	*T. brucei*	*SRA* +ve	*T. congolense*	*T. vivax*	*T. theileri*
**Trypanosome positive**	561	473	204	14	178	92	128
**Trypanosome negative**	724	94	50	0	38	23	0
**Grand Total**	1285	567	254	14	216	115	128

**Table 2 tropicalmed-05-00024-t002:** Trypanosome species identified in cattle samples. Mixed infections were a common feature.

Infection Status	*ITS* +ve	*T. b./T. c.*	*T. c./T. v.*	*T. t./T. v.*	*T. b./T. v.*	*T. b./T. c./T. v.*
**Trypanosome positive**	473	107	11	8	5	1
**Trypanosome negative**	94	6	1	0	4	3
**Total**	567	113	12	8	9	4

(T. b. = T. brucei; T. c. = T. congolense, T. v. = T. vivax; T. t. = T. theileri).

**Table 3 tropicalmed-05-00024-t003:** Results for trypanosomiasis and tick-borne-diseases (TBD) in cattle with PCV.

Infection Status	PCR						
	Total	Theil	Bab	Ana	Tryp + Any TBD	Tryp + Bab	None
**Trypanosome positive**	561	406	234	401	447	234	NA
**Trypanosome negative**	724	139	104	180	45	16	373
**Total**	1285	545	338	581	492	250	373

Theil = *Theieria*; Bab = *Babesia*; Ana = *Anaplasma*; Tryp = trypanosomes.

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
