# Peer review of "Haemoparasitic Infections in Cattle from a Trypanosoma brucei Rhodesiense Sleeping Sickness Endemic District of Eastern Uganda"

_tropicalmed, 2020, doi:10.3390/tropicalmed5010024_

Round 1
Reviewer 1 Report
The manuscript by Matovu and cols shows the prevalence of haemoparasites in cattle in a district of Uganda. The work presents an important scientific contribution, in addition to contributing to regional public policies for the control of African trypanosomiasis transmission. However, some points require minor revisions by the authors, namely:
1. Page 2 (line 64): low PCVs - This abbreviation needs to be clarified in the section "introduction".
2. Page 5 (line 199): correct the word "positive".
3. Table 2 must be presented entirely on a single page.
4. The discussion of the manuscript is complete, but the text is too dense. I suggest splitting the text into small paragraphs. For example (page 7, lines 260-285), paragraph too long.
5. Would it be possible to add some results of PCR, agarose gel with amplified products, as supplementary material to the manuscript?
Author Response
1. Page 2 (line 65): low PCVs - The abbreviation been clarified in the section "introduction" and written in full.
2. Page 5 (line 200): the word "positive" has been corrected
3. Table 2 has been presented presented on a single page 6
4. The discussion text has been split into small paragraphs. Specifically, paragraph starting on line 264 has been split into three smaller paragraphs (264-271, 272-281 and 282-290)
5. Would it be possible to add some results of PCR, agarose gel with amplified products, as supplementary material to the manuscript?
The supplementary material has been attached and comprises five (5) figures showing results of various PCRs that were run during the analysis.

Reviewer 2 Report
The paper by Matovu et al describes the prevalence of heamoparasites in cattle in a district in central Uganda that has been part of the recent outbreak of T. b. rhodesiense sleeping sickness (rHAT). This is a very well written report, employing appropriate and professional methods and discussing the findings at a high level, against the background of the aims of HAT elimination and touching on policy development towards that aim.
I have no criticism at all of this manuscript. The results are valid and clearly presented. And although the results are, in themselves, not particularly novel or surprising the value of surveys like the one here reported should be obvious to all in this field of science. Moreover, apart from genuinely underpinning the aims of HAT elimination with the statistics on incidence, the authors make a number of excellent points in the discussion, which deserve to be noted and noted well. This is particularly the case concerning the involvement of livestock holders in the elimination of rHAT zoonosis. The authors strongly highlight that interventions designed by the scientific community must be sufficiently in the interest of the farmers that they see clear benefits in participating, and call for trans-disciplinary approaches including social scientists and social economists. They also identify and highlight the problem of diminishing returns, when fewer animals carry T. b. rhodesiense but are still anaemic and otherwise ill from tick-borne diseases and helminthiases. I strongly agree with the authors that only an integrated approach will continue to drive the Tbr infections towards elimination, and that such a strategy is due now.
As such, I have only to draw attention to a few small typos, and congratulate the team on a an exemplary study and report.
Line 22. ‘Interesting’ is repeated.
Line 82. ‘Into’ not ‘in to’
Line 89. Hyphen us: I think it should be ‘trypanosome-negative’
Line 244. Trypanosome-positive. Same with tick-borne elsewhere.
Line 261. Delete ‘a’ in ‘a tick-borne parasites’.
Line 262. ‘based’ , not ‘basing’
Author Response
Line 22. ‘Interestingly' has been deleted.
Line 82. ‘into’ has replaced ‘in to’
Line 89. A hyphen has been inserted thus ‘trypanosome-negative’
Line 244. A hyphen has been inserted thus 'trypanosome-positive'.
Lines 21, 99, 261 A hyphen has been inserted thus 'tick-borne'.
Line 261. The‘a’ in ‘a tick-borne parasites’ has been deleted
Line 262. the word ‘based’ , has replaced 'basing’
